# Nephroprotective Effect of *Sansevieria trifasciata*

**DOI:** 10.3390/ijms26178619

**Published:** 2025-09-04

**Authors:** Josue Ramos Islas, Manuel López-Cabanillas Lomelí, Blanca Edelia González Martínez, Israel Ricardo Ramos Islas, Myriam Gutiérrez López, Alexandra Tijerina-Sáenz, Jesús Alberto Vázquez Rodríguez, Luis Fernando Méndez López, María Julia Verde-Star, Romario García-Ponce, David Gilberto García-Hernández, Michel Stéphane Heya

**Affiliations:** 1Facultad de Salud Pública y Nutrición, Universidad Autónoma de Nuevo León, Ave. Pedro de Alba S/N & Ave. Manuel L. Barragán, San Nicolás de los Garza 64460, NL, Mexico; josue.ramosis@uanl.edu.mx (J.R.I.); manuel.lopezcabanillaslm@uanl.edu.mx (M.L.-C.L.); blanca.gonzalezma@uanl.mx (B.E.G.M.); myriam.gutierrezl@uanl.mx (M.G.L.); alexandra.tijerinas@uanl.mx (A.T.-S.); jesus.vazquezrdz@uanl.edu.mx (J.A.V.R.); luis.mendezlop@uanl.edu.mx (L.F.M.L.); 2Facultad de Medicina, Universidad Autónoma de Nuevo León, Pedro de Alba S/N & Ave. Manuel L. Barragán, San Nicolás de los Garza 64460, NL, Mexico; drricardoramos2024@hotmail.com; 3Departamento de Química, Facultad de Ciencias Biológicas, Universidad Autónoma de Nuevo León, San Nicolás de los Garza 66455, NL, Mexico; jverdestar@gmail.com (M.J.V.-S.); qbpromario@gmail.com (R.G.-P.)

**Keywords:** kidney diseases, phytocompounds, polyphenols, nephroprotective activity, mechanism of action

## Abstract

Kidney diseases represent an increasingly significant global public health challenge, with an estimated prevalence of around 10% among adults and a rising trend influenced by factors such as population aging and exposure to nephrotoxic agents. Given the limitations of conventional treatments, which often only slow disease progression and may cause adverse effects, there is growing interest in exploring alternative therapies based on natural compounds. *Sansevieria trifasciata*, commonly known for its ornamental use, has been widely used in traditional medicine in Mexico and other tropical regions due to its antioxidant, anti-inflammatory, and regenerative properties. Recently, its phytochemical profile has drawn scientific attention, particularly due to its high content of hydroxylated aromatic compounds such as flavonoids, terpenes, and phenolic acids, which may offer protective effects on kidney function. For this review, searches were conducted in specialized databases such as PubMed, Scopus, and Google Scholar, as well as platforms like ChEMBL and SWISS, selecting articles published between 2008 and 2025. This work aims to compile and critically analyze the available scientific literature on the nephroprotective potential of the phytochemicals found in *S. trifasciata*, and includes a preliminary exploration of their possible mechanisms of action using pharmacokinetic and pharmacodynamic prediction tools.

## 1. Introduction

Kidney diseases represent an increasingly significant global health burden, with an estimated 850 million people affected by various forms of kidney damage, including both acute injuries and chronic disorders of diverse etiologies [1]. It is projected that by 2040, kidney diseases could rank among the top five causes of death worldwide [1,2], driven by the rising incidence of predisposing factors such as diabetes mellitus, hypertension, obesity, population aging, and exposure to environmental and pharmacological nephrotoxic agents [3,4]. In low- and middle-income regions, the lack of access to early diagnosis and adequate treatment further exacerbates the progression of kidney damage and worsens clinical outcomes [4]. This epidemiological trend highlights the urgent need for innovative and safe therapeutic strategies, including approaches based on bioactive plant-derived compounds with potential nephroprotective effects, such as *S. trifasciata*, whose traditional medicinal properties are beginning to be validated by scientific research.

In recent decades, plant-derived natural products have gained renewed interest as new pharmacological sources, particularly in the field of kidney disease, due to their chemical diversity and lower toxicity compared to synthetic drugs [5]. Various botanical species have demonstrated nephroprotective effects by modulating oxidative stress, inflammation, and cell apoptosis, key mechanisms in the pathophysiology of kidney damage [6]. Additionally, plants such as *Abelmoschus manihot*, *Salvia miltiorrhiza*, *Rheum* spp., *Curcuma longa*, and *Camellia sinensis*, among others, have been shown to reduce renal inflammation, proteinuria, and oxidative damage [7]. Some, like *Astragalus membranaceus* and *Curcuma longa*, have been evaluated in clinical trials, demonstrating improvement in renal function and reduction in inflammatory markers, highlighting their therapeutic potential in the management of kidney diseases [7]. In this context, *S. trifasciata*, a perennial herbaceous plant from the Asparagaceae family, has been traditionally used in ethnomedicine in Latin America, Asia, and Africa to treat inflammation, infections, and skin wounds [8,9]. Phytochemical studies have confirmed its content of pharmacologically relevant secondary metabolites, including flavonoids, triterpenoids, and phenolic acid compounds known for their ability to modulate cellular pathways involved in renal tissue protection [9,10].

Although preclinical research on the nephroprotective potential of *S. trifasciata* is still in its early stages, available data suggest beneficial effects in various experimental models of induced kidney damage. For example, several studies have proven that phytocompounds of *S. trifasciata* prevent hypoazotemic, as well as histological damage in the renal tubules of rats exposed to gentamicin, suggesting a protective effect mediated by inhibition of oxidative stress and restoration of redox balance [11,12,13,14]. Similarly, a study reported that phenolic compounds present in plant extracts, including quercetin, ferulic acid, and gallic acid, can modulate the expression of proinflammatory cytokines and antioxidant enzymes such as superoxide dismutase and catalase [15]. Furthermore, in comparative studies using cisplatin-induced nephrotoxicity models, extracts rich in flavonoids from *S. trifasciata* have been shown to reduce leukocyte infiltration and preserve the histological architecture of the renal glomerulus [9,10]. These findings, though preliminary, support the hypothesis that the plant’s phenolic metabolites act through a combination of antioxidant, anti-inflammatory, and anti-apoptotic mechanisms.

This work aims to compile the existing scientific literature on the hydroxylated aromatic compounds of *S. trifasciata*, emphasizing their nephroprotective properties. Additionally, it includes a preliminary exploration of their possible mechanisms of action using online pharmacokinetic prediction tools, with the aim of evaluating their viability as a phytotherapeutic agent for the prevention or mitigation of kidney damage and potentially other human diseases. For this purpose, information was gathered from specialized databases including PubMed, Scopus, Google Scholar, ChEMBL, and SWISS.

## 2. Predisposing Factors for Kidney Diseases

Kidney disease is an expanding global public health issue resulting from a complex multifactorial interaction of genetic, metabolic, environmental, and nutritional determinants. It is reported that over 850 million people worldwide suffer from some degree of kidney disease (KD) [1,2]. Chronic kidney disease (CKD) has become the eighth leading cause of death globally, with prevalence ranging from 8% to 16% of the adult population, depending on the region. In Latin America, incidence rates are particularly high; for instance, Mexico reports a prevalence close to 12% [1,9], which has increased the demand for replacement therapies such as dialysis and kidney transplantation. Among the main risk factors is hypertension, which, through its prolonged persistence, causes progressive damage to the renal microvasculature, promoting glomerulosclerosis and a decline in glomerular filtration rate. This relationship is bidirectional, as renal dysfunction can also exacerbate hypertension [16]. Type 2 diabetes mellitus, especially in cases of poorly controlled chronic hyperglycemia, is the leading cause of nephropathy in many countries, including Mexico, where up to 63% of dialysis patients are diabetic [9,17,18].

Obesity, particularly the accumulation of perirenal visceral adipose tissue, contributes to glomerular hyperfiltration, insulin resistance, and low-grade chronic inflammation processes that drive CKD progression [17,19,20]. Likewise, dyslipidemia promotes atherosclerotic processes in the renal vasculature, compromising blood flow and worsening structural damage [11,12,13,14,15]. Tobacco use is another significant risk factor, inducing oxidative stress and endothelial dysfunction that reduces renal perfusion. Advanced age is also associated with a progressive physiological decline in renal function, constituting a highly prevalent non-modifiable risk factor [17,19,20,21]. Beyond traditional factors, recent studies highlight the influence of environmental exposures such as pesticides and heavy metals, as well as autoimmune diseases like systemic lupus erythematosus, as potential triggers of chronic kidney damage, particularly in vulnerable populations or those with occupational exposures [22,23].

## 3. Conventional Treatment in Renal Affections

Among the conventional drugs used to treat diseases and protect against the possible effects of free radicals are, for example, angiotensin II receptor Blockers (ARBs). According to Piero Ruggenenti et al. [24] ARB treatments can alleviate overt nephropathy in type II diabetes. Similarly, angiotensin-converting enzyme inhibitors (ACE inhibitors) also provide additional renal benefits by blocking the renin–angiotensin–aldosterone system (RAAS). In contrast, ARBs can cause hypotension or renal insufficiency [25,26]. Additionally, these drugs may enhance the hypotensive effects of other antihypertensive agents, thereby increasing the risk of hypotension, acute renal failure, or hyperkalemia [25,26].

Another pharmacological group with a positive impact on kidney function includes sodium-glucose cotransporter 2 inhibitors (SGLT2 inhibitors), such as empagliflozin and dapagliflozin. Their main mechanism involves reducing the reabsorption of sodium and glucose in the kidneys, promoting osmotic diuresis that contributes to lowering intravascular volume and glomerular pressure [27]. At the renal level, SGLT2 inhibitors attenuate glomerular hyperfiltration and slow the progression of diabetic nephropathy. However, their effect is not directly aimed at oxidative stress or inflammation. Furthermore, the administration of these agents has also been associated with hypotension and increased levels of high-density lipoproteins [28].

In this context, the use of compounds derived from medicinal plants emerges as a promising and potentially reliable alternative for the prevention of kidney diseases.

## 4. Nephroprotective Effect of *Sansevieria trifasciata*

As shown in Figure 1, several studies have demonstrated the nephroprotective potential of *Sansevieria trifasciata*. This medicinal plant has proven effective in inhibiting inflammation and cellular apoptosis, as well as preventing DNA fragmentation. Additionally, it helps reduce hyperglycemia, oxidative stress, kidney stone formation, and fibrosis all of which are key contributors to renal dysfunction. At the same time, *S. trifasciata* has been shown to promote autophagy and the regeneration of kidney tissue. Altogether, these beneficial effects position *S. trifasciata* as a promising therapeutic alternative for kidney protection and functional recovery.

Although preclinical research regarding *S. trifasciata* in the realm of nephroprotection remains limited, multiple bioactive constituents within this plant, chiefly flavonoids, phenolic acids, and terpenes, have exhibited notable and promising pharmacological activities through the modulation of critical pathways implicated in renal pathologies (Table 1).

### 4.1. Flavonoids

Within the framework of phytochemical studies conducted on *Dracaena trifasciata*, FL1 ((2S)-3′,4′-methylenedioxy-5,7-dimethoxyflavone) was successfully isolated and identified as the main bioactive compound present in subfraction E, which demonstrated significant pharmacological properties [11]. This flavonoid exhibited notable antidiabetic activity through the inhibition of the α-glucosidase enzyme, standing out for greater molecular stability compared to acarbose, the standard drug used as a control, with an RMSD value of 0.49 Å, suggesting a possible prolongation of its therapeutic effect [11]. At the molecular level, the mechanism of action is based on the formation of hydrogen bonds with the Arg552 residue and hydrophobic interactions with the Asp232 residue of the target enzyme. Additionally, its capacity to inhibit cytochrome P450 isoenzymes CYP2D6 and CYP3A4 was reported, which could have relevant clinical implications in modulating the metabolism of concomitant drugs [11].

FL2 (Chrysin): Chrysin has shown significant nephroprotective effects, acting through the modulation of various molecular pathways underlying its therapeutic potential against different nephrotoxic agents [13]. Its antioxidant activity is reflected in the reduction of oxidative stress markers such as malondialdehyde (MDA), and the increase in key antioxidant enzyme activity, including reduced glutathione (GSH), catalase (CAT), superoxide dismutase (SOD), and glutathione peroxidase (GPx). Furthermore, it activates the HO-1 signaling pathway, crucial for the cellular antioxidant response [13]. FL2 exerts an anti-inflammatory effect by inhibiting nuclear factor kappa B (NF-κB) and reducing proinflammatory cytokines such as TNF-α, IL-1β, and IL-6 [13]. In terms of cellular protection, it prevents apoptosis by regulating caspase activity [3,8,9], decreases DNA fragmentation, and preserves mitochondrial integrity [13]. Animal studies have shown that administration of FL2 at doses of 25 to 50 mg/kg attenuates hypoazotemic and increases renal antioxidant levels in models of cisplatin-induced nephrotoxicity. Similar protective effects have been observed in gentamicin-induced nephrotoxicity (10–40 mg/kg) [13]. Likewise, in paracetamol-induced renal damage models, FL2 reduces TNF-α, IL-1β levels, and oxidative stress [13].

Moreover, preclinical studies have suggested that FL2 can inhibit signaling pathways related to renal fibrosis, such as SMAD and JNK/ERK pathways, showing antifibrotic activity in rat models of chronic kidney disease [31]. In diabetic nephropathy models, FL2 decreases collagen accumulation induced by advanced glycation end products (AGEs), regulates α-SMA expression, and suppresses profibrotic pathways such as TGF-β1 and SMAD2/3, thus blocking signals that promote renal fibrosis [31]. An anti-inflammatory and antioxidant effect has also been documented in adenine-induced nephropathy rats, evidenced by the decrease in inflammatory cytokines [31].

FL3 (Isoflavone): Isoflavone, a subclass of phytoestrogens, play a crucial role in suppressing the expression of biomarkers associated with inflammation and renal fibrosis, such as angiotensinogen (AGT), which contributes to renal damage in diabetic nephropathy through activation of the intrarenal renin-angiotensin system (RAS) [33]. These molecules possess potent antioxidant properties that neutralize free radicals and reactive nitrogen species, thereby protecting cell membranes from lipid peroxidation [33]. They also regulate the activity of antioxidant enzymes such as SOD, CAT, and GPx, contributing to reduced oxidative DNA damage [33]. Animal studies have shown that isoflavone administration delays the progression of diabetic nephropathy, fibrosis, and renal inflammation through activation of the Nrf2 antioxidant pathway and inhibition of profibrotic and proinflammatory signals such as TGF-β and NF-κB. These results in histological and functional improvements, including decreased creatinine, BUN, and urinary albumin-to-creatinine ratio [33].

Additionally, in vitro studies have shown that FL3 modulate inflammatory markers such as TNF-α, IL-6, IL-8, IL-1β, PGE2, and adhesion molecules, while regulating anti-inflammatory cytokines like IL-10 [34]. Regarding oxidative stress, FL3 exerts a protective effect against reactive oxygen, nitrogen, and chlorine species, which are primary contributors to inflammatory renal damage [34]. They also activate nuclear receptors such as PPAR-α and PPAR-γ, involved in cellular metabolism and regulation of glucose and lipid metabolism [34]. Experimental rat models subjected to a high-fat and fructose diet showed that isoflavone supplementation for 60 days reduces triglycerides, LDL, and glucose levels while improving renal markers such as proteinuria and creatinine clearance, decreasing inflammation and oxidative stress through PPAR-γ activation [34].

FL4 (Rutin): Is a flavonoid glycoside derived from quercetin found in various fruits and plants, and possesses well-documented antioxidant and anti-inflammatory properties [31]. Its antioxidant mechanism includes scavenging free radicals like DPPH and reducing lipid peroxidation, as well as activating the Nrf2 antioxidant pathway [31]. FL4 inhibits inflammatory pathways such as NF-κB and TGF-β1/Smad3, reducing the production of proinflammatory cytokines (TNF-α, IL-1β, IL-6) [31]. In animal models, FL4 has shown the ability to improve renal structure and reduce fibrosis after eight weeks of treatment at doses of 100 mg/kg/day, evidenced by reduced expression of type I/III/IV collagen, laminin, and fibronectin, while increasing antifibrotic proteins such as p-Smad7 and improving renal markers like urea, creatinine, and proteinuria [31]. Moreover, in diabetic nephropathy models, FL4 modulates endothelial-mesenchymal transition (EndMT), autophagy, and histone deacetylase 1 (HDAC1) expression, processes involved in renal fibrosis and endothelial dysfunction. Treatment with FL4 (200 mg/kg/day) for 8 weeks decreases proteinuria, improves glomerular filtration rate (GFR), and reduces activation of proinflammatory and profibrotic pathways, facilitating histone acetylation and promoting the expression of renal protective genes [37,38]. Its nephroprotective potential extends to reducing nephrotoxicity induced by drugs such as cisplatin, gentamicin, vancomycin, and cyclophosphamide, where it inhibits caspase activation, reduces proinflammatory cytokines, and restores renal antioxidant activity [37].

FL5 (Quercetin): A widely studied flavonoid in both traditional and modern Chinese medicine, exhibits antioxidant, anti-inflammatory, and antiapoptotic properties that contribute to preserving renal function under drug-induced nephrotoxicity or environmental contamination [15]. It neutralizes reactive oxygen species (ROS), inhibits MDA production, and activates HO-1 expression, restoring redox balance and improving renal biomarkers [15]. The combined use of FL5 and resveratrol has demonstrated reductions in p53 expression, as well as serum urea and creatinine levels, indicating a hypoazotemic effect and improvements in lipid and energy metabolism in acrylamide-induced renal damage models [15]. In acute kidney injury (AKI) models caused by ischemia–reperfusion, FL5 inhibits ferroptotic apoptosis in proximal tubular epithelial cells through its antioxidant effect [15]. Additionally, FL5 suppresses proinflammatory pathways (Mincle, Syk, NF-κB) in macrophages, modulates the inflammatory response in sepsis, and regulates autophagy in diabetic nephropathy and renal fibrosis contexts, reducing TGF-β expression and epithelial–mesenchymal transition [15]. It also improves glycemic metabolism by regulating glucose uptake and insulin levels [15].

FL6 (Kaempferol): A flavonoid found in numerous plants, has demonstrated nephroprotective, anti-inflammatory, and immunomodulatory effects relevant to chronic kidney disease [39]. Its mechanism includes inhibition of RhoA, ROCK2, and MYPT1 activation, reducing oxidative stress and the expression of inflammatory cytokines (TNF-α, IL-1β) and fibrosis markers (TGF-β1, fibronectin, collagen IV) [56]. It also regulates the Nrf2 antioxidant pathway, increasing SLC7A11 and GPX4 expression, protecting against ferroptosis in ischemia–reperfusion renal damage [56]. Furthermore, FL6 improves glycemic control by stimulating GLP-1 release and insulin secretion, enhancing its therapeutic potential in diabetic nephropathy [56].

In murine sepsis models, administration of FL6 (1 mg/kg/day) significantly improves renal function and reduces inflammatory markers and cell adhesion molecules, while modulating the adaptive immune response [39]. In cisplatin-induced nephrotoxicity, it decreases apoptosis in renal tubular cells by regulating NF-κB, AKT, and MAPK pathways [39].

FL7 (Catechin): It is a flavonoid with extensive scientific evidence, which has stood out for its efficacy in preventing urolithiasis and nephrolithiasis due to its diuretic, antispasmodic, antioxidant, and antiapoptotic properties [36]. They inhibit oxalate-induced crystal formation and reduce the activity of enzymes involved in renal damage, such as γ-glutamyltranspeptidase and N-acetyl-β-D-glucosaminidase, while increasing SOD activity and expression of antiapoptotic proteins like Bcl-2 [36]. Furthermore, FL7 modulates uric acid metabolism by inhibiting hepatic xanthine oxidase and balancing renal and intestinal absorption and excretion of this metabolite through regulation of transporters URAT1, GLUT9, OAT1, OAT3, and ABCG2, preventing hyperuricemia and associated renal damage [41]. Finally, they inhibit inflammation induced by monosodium urate crystals by blocking inflammatory pathways that contribute to nephropathy [41].

### 4.2. Phenolic Acids

PA1 (Gallic acid): Is a natural antioxidant belonging to the phenolic acid family, and has demonstrated a significant role in traditional Chinese medicine due to its biological activity against urolithiasis [12]. In vitro studies have shown that PA1 inhibits the formation of calcium oxalate (CaOx) monohydrate crystals [12]. In an experimental model using male C57BL mice, six groups were studied, including one group induced with nephrolithiasis by intraperitoneal injection of glyoxylate (75 mg/kg/day) for six days to simulate CaOx kidney stone formation. Another group received PA1 at doses of 25–50 mg/kg/day for six days, administered 10 h after glyoxylate injection. Evaluated parameters included renal crystal deposits, tubular damage markers (CD44), oxidative stress, activation of the HO-1 pathway, and renal function markers [12]. Results demonstrated that PA1 significantly improved inflammation and tubular injury, reducing CaOx deposition in renal tubules and HK-2 cells, which led to improved renal function. Activation of the Nrf2 pathway was observed, decreasing oxidative stress and cellular damage while reducing proinflammatory cytokines. Finally, PA1 suppressed CD44 and osteopontin expression, contributing to its anti-adhesive and protective effects against kidney stone formation [12].

In a diclofenac-induced nephrotoxicity model [42], the effects of PA1 were evaluated in rats divided into five groups over seven days. Diclofenac-treated animals exhibited significant increases in serum protein carbonyl, sGOT, sGPT, urea, creatinine, uric acid, nitrite, MDA, and IL-1β levels, alongside IL-1β gene overexpression. Conversely, PA1 administration mitigated diclofenac-induced oxidative stress, improved biochemical parameters, and showed histological renal improvement, highlighting its potential as an antioxidant and renal inflammation modulator [42].

PA2 (Protocatechuic acid): Acute kidney injury induced by lipopolysaccharides (LPS) activates key inflammatory mediators, including the IKBKB/NF-κB and MAPK/Erk/COX-2 pathways through Toll-like receptor 4 (TLR4), as well as oxidative stress by inhibiting total antioxidant capacity, catalase, and NAD(P)H quinone oxidoreductase 1 (NQO1) [43]. PA2 counteracts LPS-induced proinflammatory and oxidative effects by inhibiting inflammatory cascades and preserving antioxidant activity, preventing histopathological changes in renal tissue in murine models [43].

PA3 (Chlorogenic acid): This compound, present in various plants used in traditional Chinese medicine, is known for its anti-inflammatory, antioxidant, and antimicrobial properties. Although traditionally used as a “detoxifier”, its effects on renal fibrosis have been less explored. A recent study evaluated PA3 in a murine renal obstruction model and human renal cell cultures, observing a significant reduction in inflammation, renal damage, oxidative stress, and fibrosis. These effects were associated with the inhibition of the TLR4/NF-κB pathway [44], suggesting PA3 as a promising alternative for treating renal fibrosis.

PA4 (Caffeic acid): This hydroxylated aromatic compound has been primarily studied in the context of oxidative stress and renal inflammation. It protects renal tissue by inhibiting leukocyte accumulation, scavenging reactive oxygen species, and stimulating antioxidant enzymes. In diabetic nephropathy models, it reduces inflammation by decreasing proinflammatory cytokines (TNF-α, IL-6, IL-1β, MCP-1) via NF-κB pathway inhibition [31]. At the mitochondrial level, PA4 preserves outer mitochondrial membrane integrity, prevents cytochrome c release, and reduces ROS production, partially protecting mitochondrial enzymes from ischemia/reperfusion damage [45].

PA5 (Syringic acid): This hydroxylated aromatic compound has shown nephroprotective effects in murine models of diabetic nephropathy and ischemia/reperfusion injury [46]. In diabetic rats, PA5 improved renal function by reducing serum creatinine and increasing urinary creatinine and urea excretion. Molecularly, it stimulated Nrf2 expression and promoted autophagy in renal cells exposed to hyperglycemia [46]. In IRI models, it decreased oxidative stress markers (MDA, IMA, TOS, OSI) and cellular apoptosis, improving renal histology and showing substantial protective effects [47].

PA6 (Vanillic acid): Several studies have demonstrated that PA6 exerts a potent protective effect on renal function in rat models of diabetic nephropathy. Administration of PA6 at various doses significantly reduced hyperglycemia, improving renal parameters such as creatinine, urea, serum albumin, urine volume, albuminuria, and creatinine clearance [14]. At the molecular level, PA6 reduced oxidative stress and renal inflammation by decreasing proinflammatory cytokines (TNF-α, IL-1β, IL-6), TGF-β1, and NF-κB activity. It also showed superior antifibrotic effects compared to glimepiride [14]. Another study confirmed its ability to normalize renal structure and beneficially modulate inflammatory and antioxidant factors such as COX-2 [48]. Together, PA6 is a multifunctional nephroprotective agent in diabetic nephropathies.

PA7 (Ferulic acid): Exhibits significant protective potential against diabetes-associated renal damage and oxidative stress. This compound reduces inflammation, apoptosis, and enhances renal autophagy by modulating key cellular pathways involved in pathogenesis, such as advanced glycation end products (AGEs), MAPK proteins, and NF-κB [49]. It also counters excessive ROS production, promoting cellular repair. Additionally, PA7 provides cardiovascular benefits, including lowering blood pressure and cardiac damage, indirectly contributing to renal protection by increasing antioxidant enzymes like superoxide dismutase and catalase [50]. Overall, PA7 is a multifunctional agent with antihyperglycemic, antioxidant, anti-inflammatory, anti-apoptotic, and autophagy-promoting properties, making it a promising renal protector.

PA8 (Rosmarinic acid): Has shown protective effects in diabetic nephropathy models, improving tubular epithelial and podocyte damage by reducing oxidative stress and inflammation [51]. This effect is attributed to the downregulation of proinflammatory and oxidative stress-related genes such as Gas5, HIF1A, NFKB2, and STAT3, and suppression of genes involved in immune cell recruitment (Syk, Ccl9). It also modulates ROS clearance, controls immune cell infiltration, reduces natural killer (NK) cell cytotoxic activity, and decreases proinflammatory macrophages (S100a4), helping to restore a balanced immune microenvironment in affected renal tissue [51].

PA9 (Cinnamic acid): Various oxidative stress indicators were evaluated in experimental models, including antioxidant enzyme activities (catalase, superoxide dismutase, glutathione reductase, glutathione-S-transferase, glutathione peroxidase) and levels of oxidative compounds (8-hydroxy-2′-deoxyguanosine, total glutathione, malondialdehyde). Treatment with PA9 significantly improved these parameters, demonstrating a protective antioxidant effect [53].

### 4.3. Terpenes: TP1 (Citronellol)

Several studies have demonstrated that administering TP1 at doses of 50 and 100 mg/kg/day exerts a significant protective effect on the kidney against acute kidney injury (AKI) induced by rhabdomyolysis. In this context, the compound notably reduced the expression of renal damage markers such as KIM-1 and myoglobin, as well as levels of apoptosis-related proteins, including active caspase-3 and BAX [54]. These results suggest that the renoprotective mechanism of TP1 is primarily mediated by the inhibition of apoptosis, positioning this compound as a potential therapeutic agent for the treatment of rhabdomyolysis-induced AKI [54]. In an additional study, where acute kidney injury was induced by folic acid in mice, the administration of TP1 at the same doses (50 and 100 mg/kg/day) significantly improved renal function. This was evidenced by decreased serum levels of urea and creatinine, as well as reduced expression of the KIM-1 gene, a well-known marker of renal damage [55]. Furthermore, TP1 modulated the renal inflammatory response by downregulating the expression of proinflammatory genes such as NF-κB, IL-6, and IL-1β. Concurrently, it attenuated renal apoptosis by reducing levels of BAX and cleaved caspase-3, reinforcing its overall protective effect on renal tissue [55].

After analyzing the nephroprotective potential of hydroxylated aromatic compounds derived from *S. trifasciata*, it is crucial to explore how these compounds could be harnessed therapeutically. In this regard, predicting their pharmacokinetic potential becomes a key step in understanding their absorption, distribution, and metabolism in the body. Therefore, addressing the pharmacokinetic characteristics of the phenolic compounds from *S. trifasciata* could provide a more detailed insight into their viability as candidates for the development of innovative treatments.

## 5. Prediction of the Pharmacokinetic Potential of Hydroxylated Aromatic Compounds from *S. trifasciata*

In the present work, a preliminary analysis was conducted using digital platforms to obtain the pharmacokinetic profile of the hydroxylated aromatic compounds from *Sansevieria trifasciata*. Evaluar si los compuestos derivados de Sansevieria trifasciata presentan características sistémicas favorables, con potencial para alcanzar concentraciones terapéuticas eficaces en el riñón y predecir la toxicidad en otros órganos. Además, se busca analizar tempranamente su perfil metabólico para facilitar la optimización y priorización de candidatos antes de realizar ensayos específicos de nefroprotección. The objective was to evaluate whether the compounds derived from *Sansevieria trifasciata* exhibit favorable systemic characteristics, with the potential to reach effective therapeutic concentrations in the kidney and to predict toxicity in non-target organs. Additionally, the aim was to analyze their metabolic profile at an early stage to facilitate the optimization and prioritization of candidate compounds before conducting specific nephroprotective assays. For this purpose, the ChEMBL platform (https://www.ebi.ac.uk/chembl/search_results/* (accessed on 10 May 2025)) was used to generate the chemical structures in SMILES format for each molecule, and the SWISS platform (http://www.swissadme.ch/ (accessed on 10 May 2025)) was employed to predict their pharmacokinetic potential.

The results indicated that most of the hydroxylated aromatic compounds present in the plant exhibit good absorption in the gastrointestinal tract, while only two compounds (chlorogenic acid (PA3) and rosmarinic acid (PA8)) showed low absorption. These data suggest that the derivatives of *S. trifasciata* can effectively cross the physiological barrier represented by the gastrointestinal tract. Furthermore, certain hydroxylated aromatic compounds, especially flavonoids, might have the capacity to cross the blood–brain barrier (BBB), a significant challenge for many therapeutic molecules. The BBB functions as a selective filter, allowing passage only to substances that meet specific permeability criteria [57]. Additionally, some flavonoids from *S. trifasciata* may interact with transporters such as P-glycoprotein (P-gp), an efflux pump that regulates the entry of compounds into the brain by actively expelling many potentially harmful substances. Being substrates of P-gp implies that these flavonoids could be efficiently transported out of the endothelial cells of the BBB, limiting their concentration in the brain.

Therefore, this interaction suggests that these compounds could be useful in the development of treatments aimed at neurological disorders, given that their ability to cross the BBB and be modulated by P-gp may influence the cerebral distribution of certain drugs or nutrients.

Table 2 shows that the flavonoids present in *S. trifasciata* possess a notable ability to inhibit cytochrome P450 enzymes (CYP1A2, CYP2C19, CYP2C9, CYP2D6, and CYP3A4), a crucial enzymatic system involved in the metabolism of drugs and other substances in the body [58,59]. This inhibition is pharmacologically significant, as it can affect the bioavailability of various medications, altering both their therapeutic effects and toxicity profiles. Furthermore, the flavonoids from *S. trifasciata* may play a protective role in preventing chronic diseases such as cancer and renal disorders by reducing the formation of toxic metabolites and reactive oxygen species. This antioxidant and modulatory action could be key to preserving long-term renal function, particularly in contexts where drug-induced toxicity or exposure to harmful compounds poses a significant risk.

It is important to note that the results presented in this section are derived from theoretical models and simulations and, therefore, do not constitute definitive empirical evidence. While these predictions provide an initial insight into the pharmacokinetic behavior of the analyzed hydroxylated aromatic compounds, their actual applicability will depend on validation through experimental studies. Confirming these results under specific biological conditions will be essential to support their reliability.

## 6. Materials and Methods

To conduct this review, research was carried out in specialized electronic databases such as PubMed, Scopus, and Google Scholar, focusing on articles related to the nephroprotective potential of *S. trifasciata*. Complementary platforms such as ChEMBL (https://www.ebi.ac.uk/chembl/search_results/* (accessed on 10 May 2025)) and SWISS (http://www.swissadme.ch/ (accessed on 10 May 2025)) were also used to obtain the chemical structures of the phytocompounds present in this plant. A total of 59 articles published between 2008 and 2025 were selected. Various combinations of key terms were used in the search, including: (1) “nephroprotective” + “*Sansevieria trifasciata*”; (2) “antioxidant” + “*S. trifasciata*”; (3) “polyphenols” + “*S. trifasciata*”; and (4) “kidney” + “predisposing factors for disease”. It should be noted that the neuroprotective potential of the derivative of our plant of interest is sought separately. Additionally, some exploratory studies using terms such as “kidney” + “conventional treatment” were considered, in cases where the plant’s multiple properties or references to other plants were mentioned. For this work, only articles published in English and Spanish were included. Non-peer-reviewed publications, such as theses and articles lacking methodological rigor, were excluded. Priority was given to studies that, due to their relevance and quality, represented the most significant advances in the field under analysis.

## 7. Conclusions and Perspectives

The hydroxylated aromatic compounds present in *S. trifasciata*, including flavonoids, phenolic acids, and terpenes, exhibit promising nephroprotective potential. Their ability to modulate oxidative pathways, reduce oxidative stress and inflammation, as well as their probable inhibition of cytochrome P450 enzymes, suggests an active role in protecting the renal parenchyma against various toxic and metabolic insults. These effects could translate into decreased generation of reactive oxygen species, preservation of glomerular and tubular function, and reduction in tissue damage induced by nephrotoxic toxins or drugs. Although the preliminary findings are encouraging, further evidence, especially from human studies, is required to validate these effects, better understand the underlying molecular mechanisms, and assess their clinical applicability in the prevention or treatment of kidney diseases. Moreover, the formulation of these compounds in nanoparticles could represent an innovative strategy to improve their bioavailability and facilitate the translation of preclinical results into effective applications in humans.

## Figures and Tables

**Figure 1 ijms-26-08619-f001:**
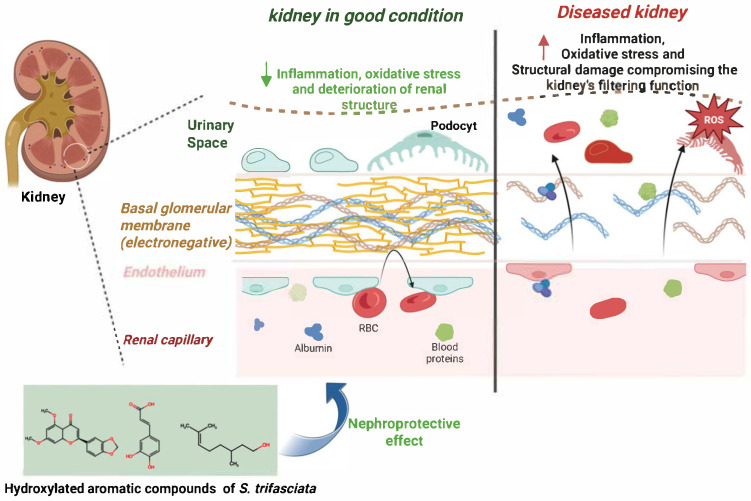
Comparative structure of a healthy kidney and one impaired by oxidative stress. In the damaged kidney, glomerular fenestrations are disrupted by reactive oxygen species (ROS), compromising filtration capacity and overall glomerular architecture. Conversely, the healthy kidney preserves structural and functional integrity. Hydroxylated aromatic compounds from *Sansevieria trifasciata* exhibit ROS-scavenging and anti-inflammatory properties, contributing to the preservation of renal ultrastructure. ↑: increase, ↓: inhibition.

**Table 1 ijms-26-08619-t001:** Possible nephroprotective potential of phenolic and alcoholic compounds derived from Sansevieria trifasciata in preclinical studies.

Group	Name	Structure	Molecular Concentration (mg/100 g of Plant)	Biological Activity
Flavonoids	(2S)-3′,4′-methylenedioxy-5,7-dimethoxyflavone (FL1)	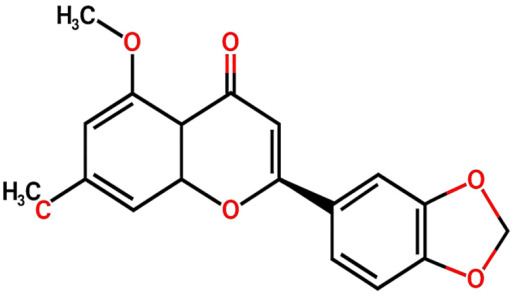	Not quantified [29]	Antidiabetic activity [11].
	Chrysin (FL2)	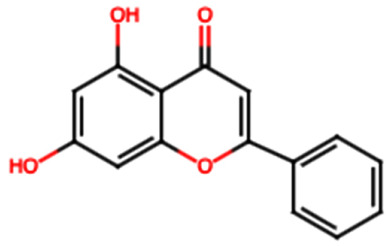	Not quantified [30]	Nephroprotective properties against various nephrotoxic agents, such as cisplatin, doxorubicin, paracetamol, gentamicin, and streptozotocin. Protection against nephropathies [13,31].
	Isoflavone (FL3)	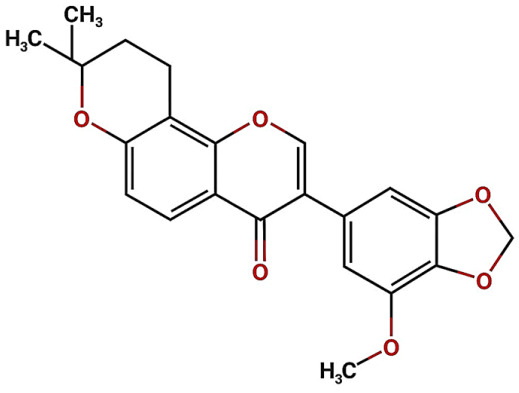	Not quantified [32]	Antioxidant activity in diabetic nephropathy. Anti-inflammatory activity in diabetic nephropathy [33,34].
	Rutin (FL4)	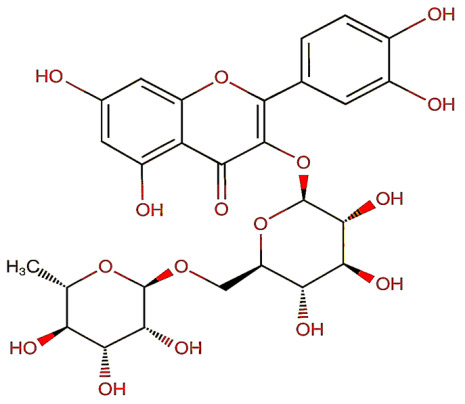	0.07 [35]	Protection against nephropathies [31,36,37,38].
	Quercetin (FL5)	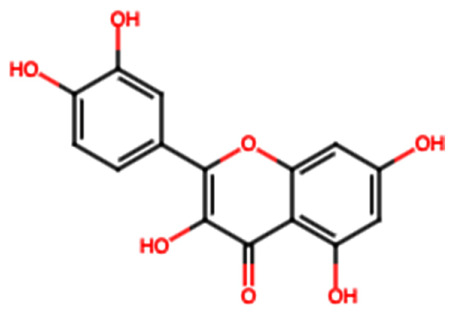	1.42 [35]	Suppresses renal toxicity, apoptosis, fibrosis, and inflammation in a variety of renal pathologies [15].
	Kaempferol (FL6)	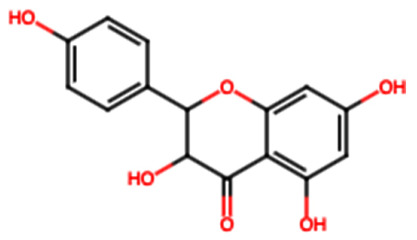	17.17 [35]	Nephroprotective effect in sepsis-induced acute kidney injury; attenuates doxorubicin-induced nephropathy [39,40].
	Catechin (FL7)	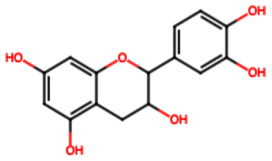	22.13 [35]	Urolithiasis prevention and anticancer activity [36,41].
Phenolic acids	Gallic acid (PA1)	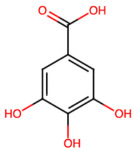	0.34 [35]	Ameliorates calcium oxalate crystal-induced renal injury and nephroprotective effect against diclofenac-induced Renal Injury [12,42].
	Protocatechuic acid (PA2)	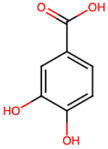	2.40 [35]	Improves lipopolysaccharide-induced kidney damage [43].
	Chlorogenic acid (PA3)	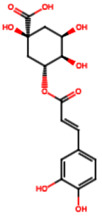	0.09 [35]	Renal antifibrotic activity [44].
	Caffeic acid (PA4)	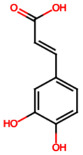	0.34 [35]	Protects the kidney against ischemia–reperfusion injury [45].
	Syringic acid (PA5)	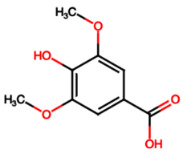	1.42 [35]	Mitigates diabetic kidney disease and protects the kidney against ischemia–reperfusion injury [46,47].
	Vanillic acid (PA6)	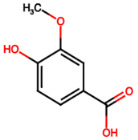	1.09 [35]	Mitigates diabetic kidney disease [14,48].
	Ferulic acid (PA7)	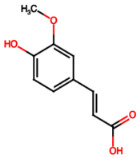	0.34 [35]	Protects against hyperglycemia-induced renal damage; improves kidney structure and function in hypertensive patients [49,50].
	Rosmarinic acid (PA8)	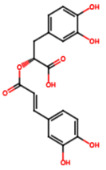	3.05 [35]	Attenuates renal tubular epithelial damage associated with diabetic nephropathy [51].
	Cinnamic acid (PA9)	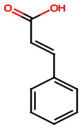	Not quantified [35,52]	Mitigates diabetic kidney disease [53].
Terpenes	Citronellol (TP1)	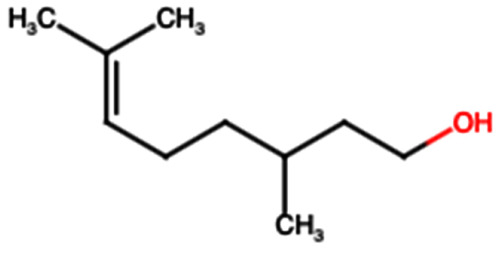	Not quantified [35,52]	Nephroprotective effects against rhabdomyolysis-induced renal injury and demonstrates anti-apoptotic activity in folic acid-induced kidney injury [54,55].

Objective of the table: This table presents relevant data on the therapeutic potential of hydroxylated aromatic compounds derived from *S. trifasciata* in preclinical studies. It includes details about the bioactive compounds, their chemical structure, their biological properties in preclinical models, highlighting their possible applicability in the development of new treatments. Note: It should be noted that all molecules mentioned have affinity with ethanol or methanol at 70%.

**Table 2 ijms-26-08619-t002:** Pharmacokinetic potential of hydroxylated aromatic compounds from *S. trifasciata* in humans.

Phytocompounds	Pharmacokinetic Targets							
	GI Absorption	BBB Permeant	P-gp Substrate	CYP1A2 Inhibitor	CYP2C19 Inhibitor	CYP2C9 Inhibitor	CYP2D6 Inhibitor	CYP3A4 Inhibitor
FL1	High	Yes	Yes	Yes	Yes	Yes	Yes	Yes
FL2	High	Yes	No	Yes	Yes	Yes	No	Yes
FL3	High	No	No	No	No	Yes	No	Yes
FL4	Low	No	Yes	No	No	No	No	No
FL5	High	No	No	Yes	No	No	Yes	Yes
FL6	High	No	No	Yes	No	No	Yes	Yes
FL7	High	No	Yes	No	No	No	No	No
PA1	High	No	No	No	No	No	No	YES
PA2	High	No	No	No	No	No	No	YES
PA3	Low	No	No	No	No	No	No	No
PA4	High	No	No	No	No	No	No	No
PA5	High	No	No	No	No	No	No	No
PA6	High	No	No	No	No	No	No	No
PA7	High	Yes	No	No	No	No	No	No
PA8	Low	No	No	No	No	No	No	No
PA9	High	Yes	No	No	No	No	No	No
TP1	High	Yes	No	No	No	No	No	No

Note: The table presents a comparative analysis of the pharmacokinetic properties of various phytocompounds, highlighting their gastrointestinal absorption (GI: high or low), ability or inability to cross the blood–brain barrier (BBB), and their interaction with key proteins such as P-glycoprotein (P-gp) and cytochrome P450 (CYP) enzymes. This type of profile is essential for predicting the efficacy and safety of compounds as potential therapeutic agents.

## Data Availability

Data is contained within the article.

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
