# Peer review of "Nephroprotective Effect of Sansevieria trifasciata"

_ijms, 2025, doi:10.3390/ijms26178619_

Round 1

Reviewer 1 Report

Comments and Suggestions for Authors

1) Considering that several times authors mention the impact of the plant or its components on creatinine and urea levels, itt would be well to state the "hypoazotemic" properties.
2) L.179-182: Animal studies have shown that administration of FL2 at doses of 25 to 50 mg/kg attenuates serum creatinine and blood urea nitrogen (BUN) 
levels in cisplatin-induced nephrotoxicity, as well as increases renal antioxidants, similarly to gentamicin (10–40 mg/kg)_ The correction is required to improve the described data.  
3) angiotensin II receptor antagonists (ARBs)_ Since abbreviation does not directly complies, it should be added synonym of the group 
(also called angiotensin II receptor blockers, or sartans)
4) Since it is uncomformably to read and revise the subchapters 4.1.-4.3, returning each time to the table, I recommend to include the substances' names within the text,
e.g. directly after the code, ie Fl1 (  (2S)-3',4'-methylenedi-oxy-5,7-di-methoxyflavone)  

Author Response

Reviewer: Considering that several times authors mention the impact of the plant or its components on creatinine and urea levels, itt would be well to state the "hypoazotemic" properties.

Authors:  The term “hypoazotemic” was used to express the potential to regulate creatinine and urea levels.

Reviewer: L.179-182: Animal studies have shown that administration of FL2 at doses of 25 to 50 mg/kg attenuates serum creatinine and blood urea nitrogen (BUN) 
levels in cisplatin-induced nephrotoxicity, as well as increases renal antioxidants, similarly to gentamicin (10–40 mg/kg) The correction is required to improve the described data.  

Authors: The writing was improved.

Reviewer: angiotensin II receptor antagonists (ARBs)_ Since abbreviation does not directly complies, it should be added synonym of the group (also called angiotensin II receptor blockers, or sartans).

Authors: The requested modification was made.

Reviewer: Since it is uncomformably to read and revise the subchapters 4.1.-4.3, returning each time to the table, I recommend to include the substances' names within the text,
e.g. directly after the code, ie Fl1 ((2S)-3',4'-methylenedi-oxy-5,7-di-methoxyflavone)  

Authors: The requested modification was made.

Reviewer 2 Report

Comments and Suggestions for Authors

In the presented review, 59 articles (published between 2008 and 2025 in English and Spanish are selected), from the available scientific literature on the nephroprotective potential of various phytocompounds: flavonoids, phenolic acids and terpenes, found in Sansevieria trifasciata were critically analyzed, including a preliminary study of their possible mechanisms of action using pharmacokinetic and pharmacodynamic prediction tools. Priority is given to research that, due to its relevance and quality, represents the most significant progress in the analyzed field.

I have a question for the authors - for what reasons was the Web of Science (WoS) electronic database not used? 

The chemical structures in Table 1 are currently unclear. The authors should provide a high-resolution version with improved formatting.

 I recommend the acceptance of the paper with small corrections.   

Author Response

Reviewer : I have a question for the authors - for what reasons was the Web of Science (WoS) electronic database not used? 

Authors: The literature search was conducted in widely recognized databases such as PubMed, Scopus, and Google Scholar, which offer extensive and up-to-date coverage of the scientific literature related to the topic. These platforms were prioritized due to their accessibility, the breadth of indexed publications, and the relevance of the results obtained for the objectives of the article. Although Web of Science (WoS) was not used, it is acknowledged as a highly prestigious and valuable database, whose inclusion could enhance the comprehensiveness of future searches.

Reviewer : The chemical structures in Table 1 are currently unclear. The authors should provide a high-resolution version with improved formatting.

Authors: The visual quality of the chemical structures was improved.

Reviewer 3 Report

Comments and Suggestions for Authors

The manuscript attempts to be a review but includes original PK/PD prediction data, which is inconsistent with the review format. Furthermore, the literature review component is not sufficiently comprehensive. The authors should either significantly expand the literature review and remove the prediction data, or reframe the paper as an original research article.

Specific Comments:

  1. The chemical structures are inconsistent in quality (e.g., Compounds 1 and 2 have background colors). Please ensure all structures are presented clearly and uniformly.
  2. The term “Phenolic Compounds” (line 382) is too general to describe all the compounds discussed. Please use more specific terminology throughout the manuscript.
  3. The rationale for predicting general metabolic PK potential, rather than parameters directly related to nephroprotection (e.g., renal excretion), is unclear. Please justify this choice.
  4. The use of bold text in Table 2 is inconsistent. Please apply a uniform formatting style.

Author Response

Reviewer: The chemical structures are inconsistent in quality (e.g., Compounds 1 and 2 have background colors). Please ensure all structures are presented clearly and uniformly.

Authors: The visual quality of the chemical structures was improved.

Reviewer: The term “Phenolic Compounds” (line 382) is too general to describe all the compounds discussed. Please use more specific terminology throughout the manuscript.

Authors: The term “phenolic compounds” was replaced with “hydroxylated aromatic compounds”.

Reviewer: The rationale for predicting general metabolic PK potential, rather than parameters directly related to nephroprotection (e.g., renal excretion), is unclear. Please justify this choice.

Authors: The selection of pharmacokinetic metabolism was specified in the text. However, the reason for predicting the overall metabolic pharmacokinetic potential, rather than focusing exclusively on parameters directly related to nephroprotection, such as renal excretion, lies in the need to understand the general systemic behavior of the compounds. Therefore, this approach provides a broader pharmacological context, which will complement the specific renal-level evaluations in a forthcoming study.

Reviewer: The use of bold text in Table 2 is inconsistent. Please apply a uniform formatting style.

Authors: The requested modification was made.

Reviewer 4 Report

Comments and Suggestions for Authors

This review aims to critically analyze the available scientific literature on the nephroprotective potential of phytochemicals from S. trifasciata. It also includes a preliminary exploration of their possible mechanisms of action using pharmacokinetic and pharmacodynamic prediction tools. While this paper is well prepared, some points must be addressed following its acceptance for publication.
1. This review merely collects data without interpreting the findings.
2. The message of this review is unclear.
3. The source of the data should be mentioned in the abstract and introduction sections.
4. The structures must be redrawn.

Author Response

Reviewer: This review merely collects data without interpreting the findings.

Authors: We sincerely appreciate your constructive comments and the positive evaluation of the manuscript’s writing quality. The present work addresses the use of Sansevieria trifasciata derivatives as green and sustainable alternatives to counteract renal disorders. As this is a review article, the main objective was to compile the existing scientific literature on the phenolic compounds of S. trifasciata, with emphasis on their nephroprotective properties, and to provide an interpretation and/or discussion of the findings. In addition, a preliminary analysis of their possible mechanisms of action was included using online human pharmacokinetic prediction tools, with the aim of strengthening their validity as phytotherapeutic agents for the prevention or mitigation of renal damage. It should be noted that the studies cited correspond exclusively to preclinical assays, which have been discussed from the perspective of the mechanism of action of each molecule analyzed in the work. Nevertheless, we acknowledge your interest in a more detailed exposition and, accordingly, we have incorporated the suggested changes into the manuscript.

Reviewer: The message of this review is unclear.

Authors: The wording of the objective of the work was improved

Reviewer: The source of the data should be mentioned in the abstract and introduction sections.

Authors: The source of the data was mentioned in the abstract and the introduction.

Reviewer: The structures must be redrawn.

Authors: The visual quality of the chemical structures was improved.

Round 2

Reviewer 3 Report

Comments and Suggestions for Authors

It can be accepted

Author Response

Thank you very much